# The Complex Connection Between Myocardial Dysfunction and Cancer Beyond Cardiotoxicity: Shared Risk Factors and Common Molecular Pathways

**DOI:** 10.3390/ijms252313185

**Published:** 2024-12-08

**Authors:** Andrea Ágnes Molnár, Kristóf Birgés, Adrienn Surman, Béla Merkely

**Affiliations:** Heart and Vascular Center, Semmelweis University, 1122 Budapest, Hungary; birgeskristof@gmail.com (K.B.); drsurmana@gmail.com (A.S.); merkely.bela@gmail.com (B.M.)

**Keywords:** myocardial dysfunction, cancer, molecular pathways

## Abstract

Cardiovascular diseases and cancer represent the largest disease burden worldwide. Previously, these two conditions were considered independent, except in terms of cardiotoxicity, which links cancer treatment to subsequent cardiovascular issues. However, recent studies suggest that there are further connections between cancer and heart disease beyond cardiotoxicity. It has been revealed that myocardial dysfunction may promote carcinogenesis, indicating that additional common pathophysiological mechanisms might be involved in the relationship between cardiology and oncology, rather than simply a connection through cardiotoxic effects. These mechanisms may include shared risk factors and common molecular pathways, such as persistent inflammation and neurohormonal activation. This review explores the connection between myocardial dysfunction and cancer, emphasizing their shared risk factors, similar biological mechanisms, and causative factors like cardiotoxicity, along with their clinical implications.

## 1. Introduction

Cardiovascular diseases and cancer represent the largest disease burden worldwide. Heart failure is a clinical syndrome resulting from structural and/or functional changes in the heart that impair its ability to fill with and pump out blood effectively [1]. The most common cause of heart failure is ischemic heart disease. However, cardiotoxicity due to cancer treatment is also a significant contributing factor [2]. Cancer is characterized by the uncontrolled growth of abnormal cells, which have unstable genomes that lead to unlimited division [3]. These cells can proliferate independently of normal growth factor regulation, replicate indefinitely, evade programmed cell death, spread to other areas, form metastases, and promote angiogenesis [3]. Heart failure and cancer were considered independent. However, as the cardiovascular and cancer-specific mortality decreases and the surviving population ages, the overlap between cardiac disease and cancer patients is increasing [4]. Patient care in this cardio-oncological population has primarily focused on the cardiotoxic side effects of oncological treatment, including myocardial dysfunction and heart failure [4]. This is a causative mechanism of cancer treatment that leads to cardiovascular disease. It represents one of the most studied fields in cardio-oncology and is defined as cancer treatment-related cardiotoxicity [2]. Nonetheless, recent publications have shown that myocardial dysfunction can also lead to concomitant or subsequent cancer, suggesting common pathophysiological mechanisms beyond the causative link between cancer and myocardial dysfunction. These may include shared risk factors and common molecular pathways, such as persistent inflammation and neurohormonal activation [4,5]. While significant progress has been made in understanding these molecular pathways, the early clinical detection of both diseases may still be difficult. Previous studies have revealed that cardiac biomarkers and imaging techniques are sensitive tools to detect early myocardial damage [2]. The 2022 European Society of Cardiology Guidelines on cardio-oncology provide guidance on the diagnosis and management of cardiovascular disease caused by cancer treatment [2]. This review discusses the link between myocardial dysfunction and cancer, highlighting common pathological pathways such as shared risk factors and similar biological mechanisms, as well as causative factors like cardiotoxicity and their clinical implications.

## 2. Common Pathological Pathways in Myocardial Dysfunction and Cancer

### 2.1. Shared Risk Factors

Epidemiological studies have shown that shared risk factors, such as aging, obesity, hyperlipidemia, smoking, hypertension, and an unhealthy diet, are associated with both cardiovascular diseases and cancer [6,7,8]. Furthermore, aging is associated with the accumulation of acquired genetic mutations, such as mutations in hematopoietic stem cells, leading to clonal hematopoiesis of indeterminate potential (CHIP) and clonal leucocytes with impaired function [9].

Hypertension is a common condition that significantly increases the risk of heart failure and cancer [6,10,11]. Recent meta-analyses have shown that hypertension is associated with a higher risk of various cancers, including kidney, breast, colorectal, endometrial, and bladder cancer. However, the mechanisms by which hypertension contributes to organ-specific cancers remain unclear [12]. Belgore and coworkers’ research found elevated plasma levels of vascular endothelial growth factor (VEGF) and its soluble receptor Fms-like tyrosine kinase-1 (Flt-1) in patients with hypertension [13]. This suggests that abnormal angiogenesis may play a role in the complications associated with hypertension by increasing blood vessel permeability, promoting cell proliferation, and influencing the migration and differentiation of endothelial cells [12,13]. Additionally, VEGF is a key factor in tumor progression [12]. Furthermore, hypertension is closely associated with the renin-angiotensin–aldosterone system (RAAS), which may also influence the risk of renal cell carcinoma [14]. Deckers and colleagues identified two single-nucleotide polymorphisms (SNPs) in the angiotensin II receptor linked to renal cell carcinoma [14]. Moreover, hypertension increases the expression and activity of matrix metalloproteinase (MMP)-2 in endothelial cells and vascular smooth muscle cells (VSMCs), due to both mechanical stress on the vascular wall and elevated levels of angiotensin II [15]. MMP-2 is essential in extracellular matrix (ECM) degradation and vascular remodeling [15]. Notably, MMP-2 and MMP-9 were upregulated in breast cancer but not in the adjacent normal tissue [16].

Obesity has been associated with both cardiovascular diseases and cancer, particularly hormone-driven cancers like breast and ovarian cancer [17,18]. The INTERHEART study revealed that truncal obesity is an independent risk factor for coronary artery disease [19]. Fatty tissue functions as an endocrine organ that produces estrogen and adipokines (e.g., adiponectin and leptin), which play a role in regulating cell growth and consequently increase the risk of hormone-driven cancers [17]. Adipose tissue produces various bioactive proteins, including pro-inflammatory cytokines such as tumor necrosis factor-alpha (TNF-α), interleukin (IL)-6, and plasminogen activator inhibitor-1 (PAI-1) [20]. These proteins activate the phosphatidylinositol-3-kinase (PI3K), mitogen-activated protein kinase (MAPK), and signal transducer and activator of transcription (STAT) 3 pathways, which are significant in atherosclerosis, angiogenesis, and tumor growth [20,21]. As a result, obesity is associated with chronic pro-inflammatory states that can lead to DNA damage and mutations, potentially contributing to cancer. Leptin dysregulation is particularly critical in carcinogenesis and metastasis. It alters the tumor microenvironment, enhances the migration of endothelial cells, and facilitates the recruitment of macrophages and monocytes, secreting VEGF and pro-inflammatory cytokines that further promote tumor angiogenesis [20]. Additionally, leptin has been found to cause endothelial dysfunction by reducing the bioavailability of nitric oxide, increasing the secretion of endothelin-1, and activating the Jun-activating kinase II (JAK)/STAT pathway, along with enhancing the extracellular signal-regulated kinase (ERK) signaling [20,22]. Furthermore, it induces the osteoblastic differentiation of vascular cells, promotes platelet aggregation, and triggers cholesterol accumulation in macrophages under hyperglycemic conditions, contributing to cardiovascular disease [20,22].

Type II diabetes mellitus is widely recognized as a significant cardiovascular risk factor. However, research has shown that diabetes may also contribute to the development of cancer, likely due to hyperglycemia [23]. Additionally, the chronic inflammation associated with diabetes could further affect carcinogenesis [23]. Insulin acts as a powerful growth factor that can promote cell growth and may induce cancer, either directly or through insulin-like growth factor-1 (IGF-1). IGF-1 is known to encourage cell proliferation and has been linked to an increased risk of colorectal cancer, prostate cancer, and premenopausal breast cancer [6,24,25]. The link between smoking, cancer, and cardiovascular disease is well known [6,26]. Active and passive cigarette smoke increases inflammation, thrombosis, oxidative stress, and the oxidation of low-density lipoprotein cholesterol, initiating atherosclerosis [27].

Genetic alterations have been associated with both cardiovascular and cancer diseases. Notable examples include missense mutations in the lipoprotein receptor-related protein 6 (LRP6) coding region located on chromosome 12p, mutations in the transcription factor 7-like 2 (TCF7L2) gene found on chromosome 10q25.3, mutations in the dual-specificity tyrosine-phosphorylation-regulated kinase 1B (DYRK1B) gene, and polymorphisms in the methylenetetrahydrofolate reductase (MTHFR) C677T gene [20]. Mutations in the LRP6 gene may lead to metabolic syndrome, coronary artery disease, and osteoporosis and can promote the development of cancers such as fibrosarcoma, hepatocellular carcinoma, breast cancer, and colorectal carcinoma [20,28,29,30]. The TCF7L2 gene acts as a transcription cofactor in the canonical wingless-related integration site (Wnt) signaling pathway. It is linked to an increased risk of several cancers, including breast, endometrial, colorectal, and recurrent prostate cancer, as well as diabetes mellitus and coronary artery disease [20,31,32,33]. The DYRK1B gene encodes a protein kinase that plays a role in cell differentiation, survival, and proliferation. This gene has been associated with hypertension, coronary artery disease, obesity, type II diabetes mellitus, and various cancers, including ovarian, lung, and pancreatic cancer [20,34]. Polymorphisms in the MTHFR gene C677T result in increased levels of plasma homocysteine, which have been associated with coronary artery disease, stroke, and several types of cancer [20,35].

### 2.2. Similar Biological Mechanisms

Various pathological mechanisms may underlie cardiovascular disease and cancer [5]. These mechanisms include inflammation, immunopathology, oxidative stress, and neurohormonal activation, which are interconnected and can lead to self-reinforcing cycles that contribute to either or both diseases [4,5].

#### 2.2.1. Inflammation and Immune System

Cardiovascular risk factors and genetic predispositions can lead to low-grade chronic inflammation, which contributes to atherosclerosis, coronary heart disease, myocardial infarction, and heart failure [5,36]. In the area affected by a heart attack, resident immune cells can recognize extracellular DNA and histones from necrotic myocardial tissue. This recognition triggers the release of pro-inflammatory cytokines by activating the nuclear factor kappa-light-chain-enhancer of activated B cells (NF-κB). NF-κB is a key regulator of the myocardial response to ischemia–reperfusion, mediating the transcription of over 150 target genes [37]. Additionally, NF-κB can activate genes involved in cell proliferation, survival, angiogenesis, and metastasis [38]. Moreover, pro-inflammatory cytokines can lead to myocardial remodeling and impair heart function, ultimately resulting in heart failure [39]. Heart failure itself also increases the levels of circulating pro-inflammatory cytokines, contributing to mild chronic systemic inflammation [40,41,42,43]. Chronic inflammation plays a critical role in the neoplastic process [44]. Inflammatory cells can induce DNA damage in proliferating cells by generating reactive oxygen and nitrogen species, resulting in genomic point mutations, deletions, or rearrangements [44]. Furthermore, pro-inflammatory cytokines inhibit p53 tumor suppressor activity, which can enhance cell proliferation [44,45]. Chronic inflammatory bowel diseases, such as ulcerative colitis and Crohn’s disease, frequently lead to colon carcinogenesis, which points out the strong relation between inflammation and cancer [44]. Notably, the potential interaction between pro-inflammatory cytokines of cardiac origin and malignant processes is still under debate. However, the CANTOS trial unveiled the crucial role of inflammation in determining cardiovascular and cancer disease [46]. The study aimed to investigate the inflammatory potential of atherothrombosis in patients with a history of myocardial infarction using a human, monoclonal anti-IL-1β antibody: canakinumab [46]. Canakinumab emerged with a significantly lower incidence of recurrent cardiovascular events and a reduced level of high-sensitivity C-reactive protein than the placebo [46]. Meanwhile, IL-1β inhibition significantly reduced the incidence and mortality of non-small-cell lung cancer (NSCLC) in the CANTOS study population [46]. It was supposed that canakinumab might enhance the effect of immune checkpoint inhibitors (ICI) and chemotherapy by inhibiting pro-tumor inflammation [47]. However, the CANOPY-1 phase III, randomized, double-blind study revealed that adding canakinumab to the first-line oncological treatment did not prolong the progression-free survival and overall survival in patients with NSCLC [47].

Notably, Meijers and coworkers proved the causal relationship between myocardial dysfunction and tumor growth in a mouse model of colon polyps [48]. SerpinA3 emerged as an acute-phase protein with pleiotropic effects on cardiac remodeling and colon tumor cell proliferation via an Akt-dependent pathway [48]. The working group highlighted that the diagnosis of heart failure may be considered a risk factor for incident cancer [48].

Immune cells play a crucial role in the tumor microenvironment by enabling cell communication and creating a supportive environment for tumor growth [49,50]. These immune cells consist of monocytes, macrophages, neutrophils, dendritic cells, and T cells, all of which also express RAAS [49]. Apart from its cardiovascular effects, it is known that the angiotensin II/type 1 angiotensin II receptor (AT1R) pathway promotes local inflammation within the tumor microenvironment by affecting immune cells, mesangial cells, and vascular smooth muscle cells [51,52]. Angiotensin II stimulates the production of the monocyte chemoattractant protein-1 (MCP-1) and its receptor, the C-C chemokine receptor 2 (CCR2), which results in the infiltration of macrophages [52,53]. M1 macrophages can inhibit tumor growth, while M2 macrophages can enhance the proliferation of cancer stem cells through the IL-6-induced activation of STAT3. This leads to the release of cytokines that support cancer stem cell renewal and the secretion of inhibitory immune checkpoint proteins by T cells, facilitating metastasis [51,54]. In summary, M2 macrophages create an environment that suppresses the immune system and supports the development of tumors, as well as the formation of new blood vessels and the invasion and spread of cancer cells [4,54,55]. Angiotensin II enhances the migration, maturation, and antigen-presenting ability of dendritic cells. It also stimulates the expression of Toll-like receptors (TLR-4) on mesangial and vascular smooth muscle cells, promoting cellular oxidative injury, apoptosis, and inflammation [52,56,57,58]. The endogenous RAAS of T cells can modify T-cell proliferation, migration, nicotinamide adenine dinucleotide phosphate (NADPH) activity, reactive oxygen species (ROS) production, and cytoskeletal rearrangement in T cells. This modification stimulates the release of cytokines and chemokines, which triggers T-cell recruitment at sites of inflammation [59,60,61].

#### 2.2.2. Oxidative Stress

Increased oxidative stress is a potentially common etiology in cardiovascular diseases and cancer. ROS are generated by mitochondrial metabolism and specific enzymes, such as NADPH oxidase, nitric oxide synthase, and xanthine oxidase [62]. It can be promoted by smoking, alcohol consumption, or radiation [62]. ROS production is harmful when it exceeds the antioxidant capacities, leading to oxidative stress and contributing to cardiovascular and cancer diseases [62]. Importantly, oxidative stress and inflammation are interconnected in both diseases [63]. Myocardial ischemia–reperfusion injury triggers the production of ROS and subsequent inflammation. This occurs through the upregulation of chemokines, the activation of neutrophil integrins, and the expression of surface adhesion molecules by endothelial cells [63,64]. Increased ROS production leads to DNA damage in cardiomyocytes, which activates the nuclear enzyme poly(ADP-ribose) polymerase 1, promoting the expression of inflammatory mediators [63]. These processes result in a subclinical inflammatory state, contributing to cardiac remodeling and heart failure [63]. Additionally, the circulating inflammatory mediator TNF-α leads to mitochondrial DNA damage and alters the activity of mitochondrial complex III, thereby increasing ROS generation [63]. Furthermore, transforming growth factor-beta (TGF-β) can also enhance mitochondrial ROS generation, facilitating ROS-mediated fibrosis in heart failure [63,65]. In conclusion, elevated ROS levels result in protein oxidation, lipid peroxidation, DNA damage, and oxidative changes in microRNAs, all of which can induce myocardial cellular dysfunction, necrosis, and apoptosis [66,67,68]. Additionally, ROS play a role in both cancer-promoting and suppressing pathways by activating transcription factors such as NF-kB, activator protein-1 (AP-1), hypoxia-inducible factor (HIF)-1α, and STAT3, while also regulating tumor suppressor genes like p53, Rb, and phosphatase and tensin homolog (PTEN) [69]. The activation of these transcription factors can lead to tumor cell proliferation, angiogenesis, and inflammation [69]. Notably, anticancer treatments achieve their effects partly through the production of ROS [69].

#### 2.2.3. Neurohormonal Activation

Sympathetic nervous system. The sympathetic nervous system plays a crucial role in regulatory functions by releasing catecholamine neurotransmitters that bind to adrenergic receptors [70]. There are two primary types of adrenergic receptors, alpha (α) and beta (β), along with various subtypes [70]. Numerous clinical and epidemiological studies have shown that chronic stress and the sympathetic nervous system can promote cancer progression through adrenergic receptors via multiple intracellular mechanisms [71,72]. Moreover, previous studies showed that cancer patients treated with β-blockers for other medical conditions tended to have lower mortality rates [73,74]. A systematic review and meta-analysis conducted by Wang J and colleagues found that β-blockade was associated with reduced cancer-specific mortality and significantly improved progression-free survival in patients with stage IV colorectal cancer [75]. Notably, most cancer types express both β1- and β2-adrenergic receptors [71]. Activating the sympathetic nervous system via β-adrenergic receptors increases the levels of 3′–5′-cyclic adenosine monophosphate (cAMP), which initiates two major downstream signaling pathways: the protein kinase A (PKA) and the MAPK pathways [71,76] (Figure 1). The activation of PKA leads to the phosphorylation of transcription factors such as cAMP response element-binding protein (CREB) in members of the GATA family. Additionally, β-adrenergic receptor kinase is recruited, inhibiting β-adrenergic receptor signaling while activating Src kinase. This sequence activates transcription factors like STAT3 and downstream kinases, leading to cellular resistance to apoptosis [76,77]. In the second major effector pathway, cAMP activates the exchange protein stimulated by adenylyl cyclase (EPAC), activating the B-Raf/MAPK signaling pathway (Figure 1). This has downstream effects on diverse cellular processes, including gene transcription, resulting in the upregulation of metastasis-associated genes involved in inflammation, angiogenesis, tissue invasion, and epithelial–mesenchymal transition (EMT), while downregulating genes that facilitate anti-tumor immune responses [76]. Additionally, PKA activates phospholipase A2 (PLA2), which releases prostaglandins and leukotrienes, further inducing cell proliferation [78]. The research by Zhang D and colleagues demonstrated that the inactivation of β1- or β2-adrenergic receptors in pancreatic cancer cells led to the inactivation of CREB, AP-1, NF-κB, and their target genes, including matrix MMP-9, MMP-2, and VEGF. This inhibition reduced cancer cell invasion, metastasis, neoangiogenesis, and proliferation [78]. Notably, sympathetic nervous system activation can occur as a consequence of heart failure, which may influence cancer progression [5]. Additionally, β2-adrenergic receptors are extensively expressed on most immune cells. They modulate antigen presentation, reduce T-cell responses, and alter type 2 immune responses [70,79]. Previous studies suggest that the upregulation of β2-adrenergic signaling can suppress anti-tumor immune responses within the tumor microenvironment [70,79]. CD8+ T cells play a crucial role in eliminating cancer cells; however, the activation of the sympathetic nervous system may lead to T-cell exhaustion through β1-adrenergic receptors [80]. Despite these findings, the understanding of the cellular and molecular mechanisms by which the β2-adrenergic pathway affects immune responses in the tumor microenvironment remains limited.

RAAS. The chronic activation of RAAS in heart failure leads to a systemic increase in the levels of angiotensin II, a peptide with pleiotropic functions [4]. This peptide is crucial in regulating blood pressure and maintaining the salt balance. During myocardial infarction, the activation of RAAS leads to the accumulation, differentiation, and migration of hematopoietic precursor cells from the bone marrow. This activation also triggers the phosphorylation of NF-κB, resulting in an inflammatory response. Prolonged RAAS activation can intensify the inflammatory response and result in the loss of cardiomyocytes. Consequently, this can lead to adverse changes in the structure of the heart, contributing to myocardial dysfunction and, ultimately, heart failure [81,82]. Previous studies have suggested that RAAS is involved in tumorigenesis, leading to a proposed association between heart failure and cancer development [4]. Increased RAAS activity has been observed in various types of cancer, including breast, kidney, pancreatic, prostate, stomach, bladder, cervix, brain, lung, liver, colon, skin, and hematopoietic cells [81].

RAAS activation involves both classical and non-classical pathways. In the classical pathway, renin converts angiotensinogen to angiotensin I, which is then converted to the effector peptide angiotensin II by the angiotensin-converting enzyme (ACE) [4]. The precursor of renin is prorenin, which was previously considered an inactive molecule until the discovery of the prorenin receptor (PRR) [83]. PRR is recognized for its crucial role in regulating RAAS [81,84] (Figure 1). It enhances the conversion of angiotensinogen into angiotensin I by boosting the activity of membrane-bound renin. This makes PRR an essential co-factor in the production of angiotensin II [81,85]. In addition to its functions in cardiovascular and renal physiology and pathophysiology, PRR has also been linked to tumorigenesis through various pathways [86]. PRR can exert cellular effects independently of angiotensin II by activating the PI3K/AKT/mammalian target of rapamycin (mTOR) and MAPK/ERK signaling pathways, thereby influencing cancer development and metastasis [85,86]. Moreover, PRR is part of the Wnt receptor complex, which also plays a role in oncogenesis without the involvement of renin [51,86,87]. It assists in binding Wnt ligands and the internalization of the receptor complex as a signalosome, thereby protecting β-catenin from inactivation [86]. Once activated, β-catenin translocates to the nucleus and binds to the transcription factor T-cell factor/lymphoid enhancer-binding factor (TCF/LEF), enhancing the expression of target oncogenes, such as cMyc, AXIN2 (which encodes axis inhibition protein 2), and CCND1 (which encodes Cyclin D1) [86]. PRR is found to be overexpressed in various types of cancer, including breast cancer, glioma, pancreatic ductal adenocarcinoma, adrenocortical cancer, prostate cancer, gastric cancer, and colorectal cancer [81,88,89,90,91,92,93,94,95].

There are two types of angiotensin II receptors: AT1R and the type 2 angiotensin II receptor (AT2R) [4,96,97]. In the classical ACE/angiotensin II/AT1R pathway, the activation of AT1R increases the aldosterone levels, promotes cell proliferation, and stimulates angiogenesis [4,81,97] (Figure 1). The non-classical pathway includes the angiotensin II/AT2R and ACE2/angiotensin 1-7/AT7R axes, which counteract the effects of the classical ACE/angiotensin II/AT1R pathway by promoting vasodilation, natriuresis, and diuresis and reducing oxidative stress through increased nitric oxide and prostaglandins [98]. The angiotensin II/AT2R pathway is linked to anti-fibrotic and anti-inflammatory effects in heart failure, as well as anti-proliferative, anti-angiogenic, and pro-apoptotic effects in cancer [4,97,99,100]. ACE2 is a metallopeptidase that is similar to ACE. It cleaves angiotensin I into the small peptide angiotensin 1-9 and converts angiotensin II into angiotensin 1-7 [98]. Neprilysin is also a metallopeptidase; it converts angiotensin I into angiotensin 1-7 and other vasoactive peptides, including kinins, endothelins, atrial natriuretic peptide (ANP), and brain natriuretic peptide (BNP) [98]. Angiotensin 1-9 activates AT2R, while angiotensin 1-7 binds to the proto-oncogene Mas receptor (MasR). This receptor has been associated with anti-fibrotic and anti-inflammatory effects in the heart, as well as anti-proliferative and anti-angiogenic effects in cancer due to a local decrease in angiotensin II levels or AT1R blockade, which results from high concentrations of angiotensin 1-7 at the tumor site [4,101] (Figure 1). The balance between the ACE/angiotensin II/AT1R pathway and the ACE2/angiotensin 1-7/MasR pathway may influence cancer development [81]. The binding of angiotensin II to AT1R activates several pathways involved in cancer development, including the PI3K/AKT/mTOR pathway, the RAS/RAF/ERK1/2 pathway, and the JAK/STAT3 pathway [81,102]. Consequently, the AT1R pathway promotes tumor cell proliferation, oxidative stress, DNA damage, hypoxia, and inflammatory processes within the tumor microenvironment [51]. Furthermore, the activation of PI3K/AKT by AT1R stimulates NF-κB, which plays a role in cell migration, increases the production of VEGF, and enhances angiogenesis and tumor growth [81,102]. Hypoxia can also stimulate angiogenesis through VEGF and upregulate ACE, HIF-1α, and HIF-2α, potentially promoting tumor progression and resistance to therapy [51]. RAAS triggers inflammation and cytokine release within the tumor microenvironment, supporting cancer cell renewal in a positive feedback loop [51].

Apart from cancer cells, RAAS is also expressed widely within other tumor cells, including epithelial cells, stromal cancer-associated fibroblasts, endothelial cells of the tumor blood vessels, and tumor-infiltrating immune cells [49,103]. As a result, it is unsurprising that paracrine RAAS’ effects play an essential role in the intercellular communication between cancer cells and their surrounding environment [49,103]. The AT1R pathway can induce EMT, facilitating cell migration and metastasis. Furthermore, AT1R can stimulate the production of cytokines that promote M2 macrophage polarization, support myeloid-derived suppressor cell maturation, and suppress the cytolytic activity of CD8+ T cells [103].

In conclusion, angiotensin II binds to AT1R, triggering processes that promote cancer development. Conversely, the angiotensin II/AT2R and angiotensin 1-7/MasR pathways support anti-cancer activity by inhibiting cell proliferation, migration, and angiogenesis [81]. It is important to note that alternative RAAS pathways can bypass ACE. For instance, cathepsins can directly convert angiotensinogen to angiotensin II, which then binds to AT1R and promotes cancer progression [51,104].

Angiotensin-converting enzyme inhibitors (ACEIs) and angiotensin II receptor blockers (ARBs) may have a role in impeding carcinogenesis [103]. These drugs were initially developed for the management of cardiovascular diseases; ACEIs work by inhibiting the synthesis of angiotensin II, while ARBs block the binding of angiotensin II to AT1R. Therefore, ARBs promote the activity of the anti-proliferative Ang 1–7/MasR and AT2R pathways, which counteract the effects of angiotensin II and the AT1R signaling [103]. Previous studies have revealed conflicting results regarding the effects of ACEIs and ARBs on cancer. However, some evidence suggests that these medications may have beneficial effects on cancer control [103,105]. Numerous studies indicate that ACEIs and ARBs can influence tumor growth by affecting tumor cells and their microenvironments, including T-lymphocyte populations, myeloid cells, tumor-associated macrophages, and cancer-associated fibroblasts.

## 3. The Causative Mechanism Between Myocardial Dysfunction and Cancer

### 3.1. Myocardial Dysfunction Triggers Subsequent Cancer

A substantial body of literature suggests that patients with heart failure have a higher risk of developing new-onset cancer and have a poorer prognosis compared to individuals without heart failure. This may be due to the combined mortality risks associated with both cancer and heart failure [106,107,108,109,110,111]. A meta-analysis conducted by Zhang and colleagues found that a history of myocardial infarction increases the risk of cancer in heart failure patients [112]. Furthermore, this meta-analysis indicated that the presence of cancer in heart failure patients leads to higher mortality rates [112]. A retrospective cohort analysis of the National Health and Nutrition Examination Survey (NHANES) data in the United States revealed that heart failure is associated with a 37% increased risk of mortality in patients without cancer and a 73% increased risk in participants with cancer, compared to those without heart failure [113]. Interestingly, conditions such as hypertension, diabetes mellitus, and coronary heart disease were not significantly associated with increased cancer mortality [113]. Chronic inflammation, neurohormonal activation, and factors released from the failing heart may stimulate tumor progression, as discussed above [6,43,48,113,114,115]. Additionally, patients with atherosclerosis face an increased risk of developing malignancies, including lung, liver, colon, and hematologic cancers [116].

Cancer progression is determined not only by genetics but also by the tumor and systemic environment [117]. Acute myocardial infarction causes pain and anxiety, which temporarily increases the sympathetic outflow from the central nervous system [118]. The heightened sympathetic activity mobilizes white blood cell progenitors from the bone marrow through beta-3 adrenergic stimulation [118]. These cells then migrate to the spleen, where pro-inflammatory monocytes are released into the circulation [118]. Monocytes are key regulators of the tumor microenvironment and have several oncogenic functions, including tumor immune evasion and angiogenesis, as well as tumor cell proliferation, migration, invasion, metastasis, and angiogenesis [118,119]. Koelwyn and colleagues discovered that myocardial infarction is a pathological stressor that can promote the growth of breast cancer in mice and humans, even in the absence of clinical signs of heart failure [120].

### 3.2. Cancer and Cancer Treatment Triggers Myocardial Dysfunction

Cancer survivors are a growing population at higher risk of developing subsequent cardiovascular disease. This increased risk stems from shared risk factors, the biological processes discussed earlier, and the cardiotoxic effects of cancer treatments [121]. A large population-based study involving 18714 participants, with an average follow-up period of 12 years, found that cancer survivors had a higher cardiovascular risk compared to individuals without a cancer history. One-third of cancer survivors went on to develop cardiovascular disease, with the highest rates observed in those with lung and hematological cancers [121]. The most common types of cardiovascular disease among these survivors were ischemic heart disease, arrhythmia, and heart failure [121]. Traditional cancer treatments like anthracyclines and radiation have been connected to dose-dependent cardiotoxicity, including symptoms of heart failure. Meanwhile, radiation therapy, particularly when targeting the chest, has been linked to harmful effects on the myocardium, valves, pericardium, and blood vessels. Over the past few decades, cardio-oncology has advanced due to the increase in cancer treatments and the associated risk of heart-related complications. A deeper understanding of the molecular pathways has resulted in the development of more targeted and selective cancer therapies (such as human epidermal growth factor receptor (HER) 2 inhibitors and VEGF signaling pathway inhibitors), multitargeted tyrosine kinase inhibitors, immunomodulatory drugs, proteasome inhibitors, and ICIs [122]. As a result, the long-term survival rates have improved, leading to a higher incidence of cardiovascular disease related to cancer therapies [123].

The 2022 ESC Guidelines on cardio-oncology recommend using the term cancer therapy-related cardiovascular toxicity (CTR-CVT) to refer to conditions such as cardiomyopathy, heart failure, myocarditis, vascular toxicity, hypertension, cardiac arrhythmia, a prolonged corrected QT interval (QTc), and pericardial and valvular heart diseases resulting from cancer treatment, including chemotherapy, targeted agents, immune therapies, and radiation therapy [2]. Cancer therapy-related cardiotoxicities significantly contribute to the development of cardiomyopathy [123]. In addition, myocardial dysfunction can also be associated with cardiac light chain (AL) amyloidosis, cardiac metastases, accelerated atherosclerosis, stress cardiomyopathy, and systemic and pulmonary hypertension [123]. The prevalence of myocardial dysfunction in cancer survivors is higher compared to individuals without malignancy. A study conducted in Rochester, Minnesota found that survivors of breast cancer and lymphoma were three times more likely to develop new-onset heart failure compared to individuals without these conditions. This increased risk was associated with anthracycline treatment. Importantly, the risk remained elevated even after accounting for various factors, including age, sex, diabetes, hypertension, coronary artery disease, high cholesterol, obesity, and tobacco use [124]. This review briefly discusses the most significant cardiotoxic agents in traditional and targeted cancer treatment (Table 1).

Anthracyclines. In cancer cells, the primary cellular target of anthracyclines is topoisomerase II-alpha (TOPO-IIα), an enzyme that cleaves DNA strands and generates transient double-strand breaks [125]. Anthracyclines bind to and stabilize TOPO-IIα–DNA cleavable complexes, leading to DNA double-strand breaks that result in programmed cell death [126]. However, in cardiomyocytes, anthracyclines bind to topoisomerase II-beta (TOPO-IIβ), leading to permanent DNA double-strand breaks and thus activating the DNA damage response. This process involves multiple pathways, including the p53 signaling pathway, which initiates an apoptosis cascade [126]. Anthracyclines disrupt nuclear DNA and can cause mutations and mitochondrial defects. They can also bind to cardiolipin, a crucial lipid in the inner mitochondrial membrane, leading to impaired oxidative phosphorylation [127]. Anthracyclines can also cause changes in mitochondria, such as the accumulation of mitochondrial iron, lipid peroxidation, protein nitrosylation, and abnormalities in calcium handling [126]. Despite the significant impact of anthracyclines on cancer treatment, cardiotoxicity remains a significant challenge in clinical routines [126]. Among all drugs known to cause severe cardiotoxicity, anthracyclines, the oldest chemotherapeutic drugs, are still widely used in the treatment of solid and hematological tumors [127]. In a meta-analysis of 22,815 cancer patients treated with anthracyclines, 17.9% exhibited early signs of cardiotoxicity, 6.3% developed clinically relevant cardiotoxicity, and 10.9% experienced cardiac events [128]. Anthracycline-induced cardiotoxicity is a dose-dependent and cumulative process that can lead to various issues, such as myocardial dysfunction, arrhythmias, and, rarely, acute myocarditis [126,127]. It can be diagnosed in up to 20% of all patients receiving anthracyclines and 48% treated with high doses of anthracyclines [129]. Combining anthracycline therapy with radiotherapy and/or monoclonal antibodies can worsen toxicity [127]. This process starts at the level of myocardial cells and gradually progresses to heart failure [126,130,131]. Acute complications are rare and occur during treatment or a few weeks later, mainly as arrhythmias [127,132]. However, chronic complications are more significant and characterized mainly by asymptomatic or symptomatic left ventricular systolic dysfunction in the early (within one year after treatment) or late stages [127,132]. It can eventually progress to dilated cardiomyopathy and congestive heart failure [127].

Fluoropyrimidines. Fluoropyrimidines, such as 5-fluorouracil (5-FU) and capecitabine, have been widely used as traditional chemotherapeutic agents for the treatment of head, neck, and gastrointestinal tumors for over half a century [133,134]. However, 5-FU is the second most common drug associated with cardiotoxicity after anthracyclines [133]. It is linked to myocardial ischemia and heart failure [132,135,136]. Less commonly, it can lead to arrhythmias, myocarditis, pericarditis, and Takotsubo cardiomyopathy [133,137]. Capecitabine, an orally administered chemotherapeutic agent, is metabolized at the tumor site to 5-FU and is thought to have less significant cardiac toxicity. In a systematic review and meta-analysis involving 63,186 patients, the incidence of fluoropyrimidine-associated cardiotoxicity was 5.04%, with myocardial ischemia (2.24%) and arrhythmia (1.85%) being the most frequent [134]. In a retrospective analysis from the CAIRO studies and Dutch Colorectal Cancer Group, the incidence of capecitabine-related cardiotoxicity in metastatic colorectal cancer was found to be 5.9% of patients, with severe cardiotoxicity in 2.3% of patients [138]. The two primary mechanisms of 5-FU-related cardiotoxicity are myocardial ischemia, mainly due to coronary vasospasm, and the direct toxic effect on cardiomyocytes [133]. Vasospasm can be related to endothelial and smooth muscle cell dysfunction, affecting the coronary macrovasculature and microvasculature [133]. Epicardial coronary artery vasospasm is usually observed in a single vessel supplying the largest territory of the myocardium and manifests in segmental myocardial dysfunction. In contrast, diffuse myocardial dysfunction can be detected in the vasospasm of the microvasculature [133]. The drug activates protein kinase C, leading to the vasospasm of the coronary arteries. Additionally, it can directly damage vascular endothelial cells, leading to microthrombosis [132]. The pathomechanism of 5-FU-induced direct cardiomyocyte and endothelial damage leads to apoptosis related to its metabolite (fluoroacetate). This mechanism differs from the effect on neoplastic cells [133,139]. Furthermore, 5-FU can lead to cellular damage, either by reduced aerobic efficacy as a consequence of mitochondrial uncoupling or by oxygen stress due to increased levels of ROS (such as superoxide anions) or the diminished activity of antioxidants (such as sodium oxide dismutase and glutathione peroxidase) [133,139,140]. Refaie and coworkers found that fluorouracil elevated the levels of cardiac enzymes, tissue malondialdehyde (MDA), IL-6, STAT4, and caspase-3 and reduced glutathione (GSH), total antioxidant capacity, and peroxisome proliferator-activated receptor alpha (PPARα) expression [141]. Moreover, 5-FU has been associated with transient myocardial dysfunction, leading to apical ballooning and the hyperdynamic contraction of the basal segments in the absence of obstructive coronary artery disease, defined as chemotherapy-induced Takotsubo cardiomyopathy [133].

HER2-targeted therapies. HER2-targeted therapies are essential in treating patients with HER2-positive invasive breast cancer and HER2-overexpressing metastatic gastric adenocarcinomas [2]. HER2 (also known as ErbB2) is a member of the HER family, which also includes ErbB1 (epidermal growth factor receptor (EGFR)), ErbB3, and ErbB4. The most commonly used HER2-targeted drugs are trastuzumab, lapatinib, and neratinib [132]. Trastuzumab is a monoclonal antibody that specifically binds to HER2, inhibiting its downstream signaling and activating cell-mediated cytotoxicity. Lapatinib and neratinib are tyrosine kinase inhibitors that compete with intracellular ATP to block HER2 signaling. This action prevents phosphorylation and downstream molecular pathway alterations, exerting anti-tumor effects [132]. Anti-HER2 therapies may cause left ventricular dysfunction in 15–20% of patients [142,143]. In breast cancer trials, trastuzumab led to symptomatic heart failure in 2 to 4% of patients, and the incidence of cardiac dysfunction was 3 to 19% [144,145,146]. Most patients treated with trastuzumab who develop cardiomyopathy experience an improvement in their clinical or cardiac function after the cessation of the treatment. However, approximately one-third of these patients still have some degree of persistent cardiac dysfunction [122,144]. HER2 inhibitors can impact multiple pathways, including neuregulin 1 (NRG1), oxidative stress, and ferroptosis, leading to cardiotoxicity. NRG1 is a regeneration growth factor released by endothelial cells. It is responsible for cardiac development and protection from stress [147,148]. NRG1 promotes myocardial repair through the stimulation of cardiomyocyte proliferation, stem cell recruitment, angiogenesis, and extracellular matrix remodeling [148,149,150]. Trastuzumab interferes with the NRG1–ErbB4–ErbB2 axis in the myocardium and inhibits the MAPK and PI3K pathways, leading to myocardial injury [132,151]. HER2 inhibitors may also cause cardiotoxicity by disrupting the intracellular antioxidant system, increasing mitochondrial ROS production [152].

VEGF inhibitors. Inhibitors that target the VEGF signaling pathway include VEGFA monoclonal antibodies, VEGF receptor 2 (VEGFR2) monoclonal antibodies, and tyrosine kinase inhibitors with anti-VEGF activity. These inhibitors are used to treat various types of cancer, such as renal, thyroid, and hepatocellular carcinomas. It is important to note that tyrosine kinases are critical in cardiovascular homeostasis, including vascular, metabolic, and myocardial regulation [153]. Therefore, inhibiting these kinases may result in a wide range of cardiovascular issues, including hypertension, heart failure, myocardial infarction, QTc prolongation, and acute vascular events [2,154,155,156]. A meta-analysis of 10,647 patients found that 2.39% of patients experienced heart failure [157]. In a small study involving 40 patients who received VEGF inhibitors, 8% of the patients developed clinically asymptomatic cancer therapeutic-related cardiac dysfunction, and 30% developed clinically significant decreases in global longitudinal strain, which is a marker for early subclinical myocardial dysfunction [158].

Multitargeted tyrosine kinase inhibitors. Tyrosine kinase inhibitors that target BCR-ABL1 are the primary treatment for chronic myeloid leukemia (CML). A recent study examined the occurrence of cardiovascular side effects in 531 patients who were treated with first-line tyrosine kinase inhibitors. The findings revealed that 45% of patients experienced cardiovascular adverse events, with 9% experiencing atherothrombotic events and 33% experiencing hypertension [159]. Tyrosine kinase inhibitors can induce cardiomyocyte apoptosis by inhibiting pro-survival pathways (AKT and ERK) and upregulating pro-apoptotic pathways (Bax, Bcl-xL, and caspase) [160].

Proteasome inhibitors. Proteasome inhibitors have been shown to reduce tumor cell invasion and metastasis, thereby slowing down the progression of malignant tumors. However, it is important to note that these inhibitors can also lead to adverse cardiac effects, such as heart failure, atherosclerosis, myocardial infarction, and cardiac arrest [132,161]. According to a meta-analysis, the incidence of high-grade cardiotoxicity associated with bortezomib was 2.3% [132,162]. The development of proteasome inhibitor-associated cardiotoxicity is primarily linked to the dysregulation of calcium ion homeostasis and abnormal energy metabolism in cardiomyocytes, such as a decrease in ATP synthesis, leading to reduced cardiomyocyte contractility, or increasing protein phosphatase 2A (PP2A) activity and inhibiting adenosine monophosphate-activated protein kinase-alpha (AMPKα), leading to a reduction in left ventricular function [132,163,164].

Taxanes. Taxanes, including paclitaxel, docetaxel, and cabazitaxel, work by affecting the tubulin proteins. This leads to the dysfunction of microtubules, which in turn inhibits cell division [165]. They are used to treat ovarian, breast, non-small-cell lung, Kaposi’s sarcoma, prostate, stomach, and head and neck cancers [165]. However, taxanes might lead to arrhythmias due to abnormal calcium ion concentrations, abnormal energy metabolism, cardiomyocyte damage, and apoptosis [165,166].

Immune checkpoint inhibitors. Cytotoxic T-lymphocyte-associated protein 4 (CTLA-4) and programmed cell death protein-1 (PD-1) are proteins that regulate the activity of T cells and play a role in preventing autoimmune diseases. Nonetheless, they can also hinder the immune system’s ability to fight cancer cells [167,168]. Monoclonal antibodies targeting these proteins, known as immune checkpoint inhibitors, are used in cancer treatment. It is believed that the use of immune checkpoint inhibitors may lead to an increase in the autoimmune response, resulting in myocarditis [167,168]. While the incidence of ICI-associated myocarditis is low, the mortality rate is high. In a multicenter registry, the prevalence of myocarditis was 1.14%, with a median onset time of 34 days after starting therapy [169]. However, the true incidence of subclinical or smoldering myocarditis might be even higher as the troponin level is elevated by 10% [170]. A combination of ICI therapy, hypertension, diabetes, smoking, obesity, and pre-existing autoimmune disease may be a risk factor for ICI-associated myocarditis [171,172,173].

**Table 1 ijms-25-13185-t001:** Brief overview of the most significant cardiotoxic agents in traditional and targeted cancer treatment.

Class	Drugs	Potential Cardiotoxicity
** *TRADITIONAL CANCER THERAPIES* **		
Anthracyclines	doxorubicin, epirubicin	cardiomyopathy, arrhythmia, myocarditis, pericarditis [128]
Platinum	cisplatin, oxaliplatin	myocardial ischemia [174,175]
Antimetabolites	fluorouracil, capecitabine	coronary spasms, myocardial ischemia, arrhythmias [134,138]
Alkylating agents	cyclophosphamide	heart failure, myocarditis, pericarditis [176]
Antimicrotubule agents	paclitaxel, vinca alkaloids	arrhythmias, myocardial ischemia [166]
**TARGETED CANCER THERAPIES**		
HER2 inhibitors	trastuzumab, pertuzumab	cardiomyopathy [144,145,146]
VEGF signaling pathway inhibitors	bevacizumab, sunitinib	hypertension, cardiomyopathy, thromboembolic events (arterial or venous) [154,155,156,157,158]
Multitargeted tyrosine kinase inhibitors	dasatinib, ponatinib, ibrutinib, trametinib	hypertension, atherothrombotic cardiovascular events, cardiomyopathy, arrhythmias [159,177,178]
Immunomodulatory drugs	thalidomide, lenalidomide	thromboembolic events, arrhythmia [179]
Proteasome inhibitors	bortezomib, carfilzomib	cardiomyopathy, hypertension, thromboembolic events (arterial and venous) [162]
Immune checkpoint inhibitors	pembrolizumab, nivolumab	myocarditis [169]

## 4. Clinical Aspects of Myocardial Dysfunction in Cancer Patients

The 2022 ESC Guidelines on cardio-oncology recommend using the term cancer therapy-related cardiac dysfunction (CTRCD) to encompass cardiac injury, cardiomyopathy, and heart failure [2]. The clinical assessment of CTRCD relies on symptoms, cardiac biomarkers, and left ventricular function [2,180,181]. It is divided into clinically asymptomatic and symptomatic cardiotoxicity, from very severe to mild grades [2]. Left ventricular dysfunction is routinely assessed by transthoracic echocardiography, which includes two-dimensional or three-dimensional left ventricular ejection fraction (LVEF) and two-dimensional left ventricular global longitudinal strain measurement (LVGLS) [180,181] (Figure 2). LVEF is the fraction of the chamber volume ejected in systole divided by the left ventricular end-diastolic volume; however, it is affected by the loading conditions and ventricular geometry. LVGLS measures the left ventricular wall deformation in the tangential base-to-apex direction, which defines left ventricular systolic shortening [182]. It is a more sensitive imaging tool for the detection of subtle systolic dysfunction than the ejection fraction, as it can partially overcome the limitations associated with ejection fraction measurement [182]. Consequently, LVGLS can detect myocardial dysfunction earlier than the conventional ejection fraction measurement. Previous studies have shown that LVGLS can effectively detect early subclinical cardiac dysfunction in patients treated with anthracyclines or HER2, VEGF signaling pathway, and immune checkpoint inhibitors [183,184,185]. Additionally, the global longitudinal strain (GLS) is valuable in demonstrating the temporal changes in early myocardial damage caused by anthracyclines and HER2 inhibitors. This information may assist in ensuring the prevention of late cardiovascular events [184]. Therefore, systolic myocardial strain analysis should be performed whenever available to detect cardiotoxicity as early as possible [2]. Even in asymptomatic cases, a new LVEF reduction to <40% is defined as severe CTRCD, whereas moderate CTRCD is described as at least a 10% reduction in LVEF to 40–49% or a less than 10% LVEF reduction with a new relative decline in LVGLS by >15% from baseline or a new elevation in cardiac biomarkers [2]. The most commonly used cardiac biomarkers are natriuretic peptides, such as BNP or *N*-terminal prohormone of BNP (NT-proBNP) and troponin. These biomarkers detect early myocardial injury, particularly in patients undergoing treatment with anthracyclines and/or trastuzumab [2]. Atrial and ventricular cardiomyocytes release BNP and NT-proBNP in response to increased myocardial wall stress in patients with myocardial dysfunction [36]. Troponin is a protein in thin myocardial filaments that activates myocardial contraction when calcium binds. Both natriuretic peptides and troponin are of prognostic value in cardiac diseases. Notably, the 2022 ESC Guidelines on cardio-oncology recommend evaluating not only the left ventricular systolic function and cardiac biomarkers but also the diastolic function, left ventricular filling pressures, right ventricular function, and ventricular diameters [2].

The LVEF is primarily responsible for determining the initiation of cardioprotective therapy in patients at risk of CTRCD [186]. The SUCCOUR (Strain Surveillance of Chemotherapy for Improving Cardiovascular Outcomes) study aimed to determine whether using LVGLS-guided cardioprotection could prevent a decrease in LVEF and the development of CTRCD in high-risk patients undergoing potentially cardiotoxic chemotherapy, as compared to standard care. The study found the lower occurrence of significant LVEF reductions in the LVGLS-guided group at the one-year follow-up. Data from the 3-year follow-up showed an even greater improvement in LVEF than at the 1-year follow-up; however, there was no noticeable difference between the LVGLS-guided and LVEF-guided groups [186,187].

Cardiovascular surveillance in cancer patients is emphasized in the 2022 ESC Guidelines on cardio-oncology and was reviewed by Gao and colleagues [2,132]. The guidelines outline the assessment of cardiovascular risk factors and myocardial function before and during cancer treatment to identify cardiotoxicity promptly [2]. The long-term follow-up of cancer survivors after treatment is recommended to detect and manage potential late cardiotoxicity [2]. If moderate or severe symptomatic anthracycline-induced cardiotoxicity is observed, chemotherapy should be temporarily interrupted [2,132]. Similarly, HER2 inhibitor chemotherapy should be interrupted in symptomatic or asymptomatic cases with a decrease in LVEF to <40% [2,132]. In asymptomatic moderate or mild HER2 inhibitor-induced cardiotoxicity, chemotherapy can be administered by ACEI/ARB and/or β-blockers [2,132]. Meanwhile, 5-FU chemotherapy should be stopped as soon as symptoms develop, and calcium channel blockers or nitrates should be used empirically, as they have been shown to improve coronary artery spasms significantly [2,132]. Daily home blood pressure monitoring is recommended during the first VEGF inhibitor cycle and every 2–3 weeks [2,132]. Patients with blood pressure ≥140/90 mmHg should be treated according to the guidelines [2,132]. The 2022 ESC Guidelines recommend baseline electrocardiography in patients treated with second-generation BCR-ABL tyrosine kinase inhibitors and the discontinuation of chemotherapy if pulmonary hypertension develops [2,132]. Electrocardiography and cardiac biomarkers are indicated before and during ICI therapy. In suspected ICI-associated myocarditis, it is recommended to perform transthoracic echocardiography and cardiac magnetic resonance, and patients should be immediately withdrawn from ICI therapy [2,132]. For hemodynamically unstable patients, high-dose methylprednisolone is administered intravenously [2,132].

## 5. Conclusions

Cardiovascular disease and cancer are the leading causes of death worldwide. Both conditions share essential modifiable risk factors, including diet, a sedentary lifestyle, obesity, and tobacco use. Additionally, non-modifiable factors, such as inflammation, significantly contribute to their development [20]. While cardiovascular disease and cancer are diverse clinical conditions, they exhibit overlapping biological mechanisms, including common genetic, cellular, and signaling pathways. Understanding these pathways could help to identify new therapeutic and preventive strategies for both diseases [20]. Moreover, patients with cardiovascular diseases, particularly those with atherosclerosis, face an increased risk of developing malignancies, such as lung, liver, colon, and hematologic cancers [116]. Additionally, cancer survivors represent a growing population that is at an increased risk for subsequent cardiovascular disease due to the cardiotoxic effects of cancer therapies, biological processes related to cancer, and shared risk factors [121].

## 6. Limitation

A search strategy using MeSH terms with defined inclusion and exclusion criteria was not feasible, which introduced potential bias.

## Figures and Tables

**Figure 1 ijms-25-13185-f001:**
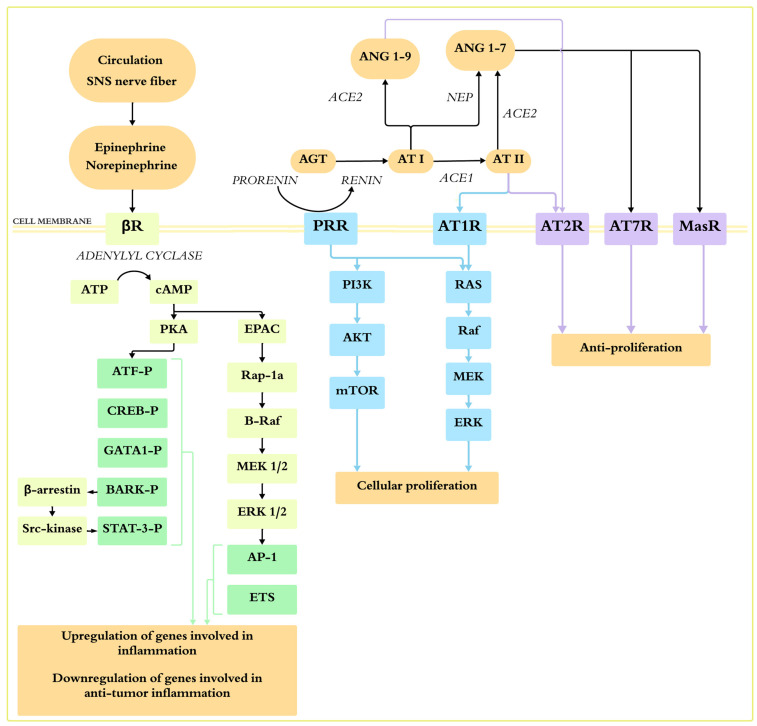
Schematic illustration showing the effects of the sympathetic nervous system (SNS) and the renin–angiotensin system (RAS) on cancer development. The binding of epinephrine and norepinephrine to the β-adrenergic receptor leads to the activation of adenylyl cyclase and the conversion of ATP into cAMP, which activates two major pathways. In the first pathway, the PKA pathway phosphorylates multiple target proteins, including transcription factors of the CREB/ATF and GATA families and β-adrenergic receptor kinase (BARK). The phosphorylation of BARK leads to the recruitment of β-arrestin, which activates Src kinase, resulting in the activation of transcription factors such as STAT3. In the second pathway, the cAMP activation of the exchange protein activated by adenylyl cyclase (EPAC) leads to the Rap1A-mediated stimulation of the B-Raf/mitogen-activated protein kinase signaling pathway, affecting gene transcription. Overall, β-adrenergic signaling results in the upregulation of genes involved in inflammation and the downregulation of genes involved in anti-tumor immune responses. The RAS pathway interacts with two main downstream routes, the Ras/RAF/MEK/ERK pathway and the PI3K/AKT/mTOR pathway, both promoting cellular proliferation. Angiotensin II binds to the AT1 receptor (AT1R), initiating processes that promote cancer development. In contrast, the angiotensin II/AT2 receptor (AT2R) and angiotensin 1-7/Mas receptor (MasR) pathways support anti-cancer activity by inhibiting cell proliferation.

**Figure 2 ijms-25-13185-f002:**
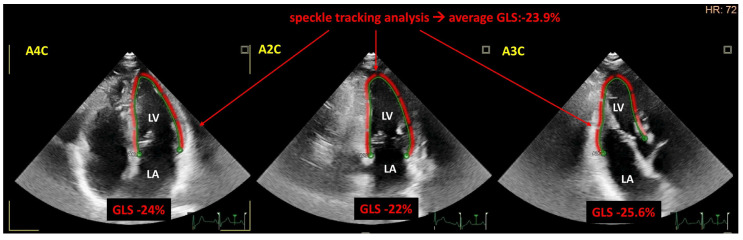
Representative image of left ventricular strain analysis using two-dimensional speckle tracking echocardiography in the case of a normal subject receiving anthracyclines. The absolute values of the left ventricular global longitudinal strain (GLS) obtained from the apical four-chamber view (A4C), two-chamber view (A2C), and three-chamber view (A3C) are within the normal range. LA: left atrium, LV: left ventricle.

## Data Availability

Not applicable.

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
