# Peer review of "The Complex Connection Between Myocardial Dysfunction and Cancer Beyond Cardiotoxicity: Shared Risk Factors and Common Molecular Pathways"

_ijms, 2024, doi:10.3390/ijms252313185_

Round 1
Reviewer 1 Report
Comments and Suggestions for Authors
In this interesting paper, the Authors described the main risk factors for both cardiovascular disease and cancer: hypertension, obesity, type 2 diabetes and genetic alterations. They also reported several pathological mechanisms underpinning both cardiovascular disease and cancer: inflammation, oxidative stress and neuro-hormonal activation. Moreover, they discussed the causative mechanisms between myocardial dysfunction and cancer. Importantly, patients with heart failure have an increased risk of developing cancer and anti-cancer treatment may facilitate myocardial dysfunction due to a potential cardiotoxic effect. With this regards, the authors exhaustively described the cardiotoxicity related to anthracyclines, fluorpyrimidines, anti-HER2 agents, ICI, taxanes and VEGF inhibitors. Finally the authors focused on the most recent guidelines, describing the clinical characteristics of cancer therapy-related cardiac dysfunction. Speckle tracking echocardiography is an innovative echocardiographic method that allows the clinicians to early detect subclinical myocardial dysfunction, defined as LV-GLS impairment in the presence of preserved LVEF (>50%) and in absence of clinical symptoms. In the paragraph 4. Clinical aspects of myocardial dysfunction in cancer patients, the concept of subclinical myocardial dysfunction could be better emphasized by the authors. The authors could mention and discuss the following references: PMID: 36620612, PMID: 36522712 and PMID: 32519318.
Author Response
|
Response to Reviewer #1
|
||
|
We thank Reviewer#1 for carefully evaluating the manuscript and for the helpful and constructive suggestions. We have heeded the Reviewer's helpful proposition and prepared a revised version of the manuscript, which includes the alterations suggested by the Reviewer. According to the Reviewer's suggestion, we have completed paragraph 4 to emphasize the concept of subclinical myocardial dysfunction better, as highlighted below:
“4. Clinical aspects of myocardial dysfunction in cancer patients The 2022 ESC Guidelines on cardio-oncology recommend using the term cancer therapy-related cardiac dysfunction (CTRCD) to encompass cardiac injury, cardiomyopathy, and heart failure [2]. Clinical assessment of CTRCD relies on symptoms, cardiac biomarkers, and left ventricular function. [2,180,181]. It is divided into clinically asymptomatic and symptomatic cardiotoxicity, from very severe to mild grade [2]. Left ventricular dysfunction is routinely assessed by transthoracic echocardiography, which includes two-dimensional or three-dimensional left ventricular ejection fraction (LVEF) and two-dimensional left ventricular global longitudinal strain measurement (LVGLS) [180,181] (Figure 2). LVEF is the fraction of chamber volume ejected in systole divided by left ventricular end-diastolic volume; however, it is affected by loading conditions and ventricular geometry. The LVGLS measures left ventricular wall deformation in a tangential base-to-apex direction, which defines left ventricular systolic shortening [182]. It is a more sensitive imaging tool for detecting subtle systolic dysfunction than ejection fraction, as it can partially overcome the limitations associated with ejection fraction measurement [182]. Consequently, LVGLS can detect myocardial dysfunction earlier than the conventional ejection fraction measurement. Previous studies have shown that LVGLS can effectively detect early subclinical cardiac dysfunction in patients treated with anthracyclines, HER2, VEGF signaling pathway, and immune checkpoint inhibitors [183-185]. Additionally, global longitudinal strain (GLS) is valuable in demonstrating the temporal changes in early myocardial damage caused by anthracyclines and HER2 inhibitors. This information may assist in stratifying the prevention of late cardiovascular events [184].”
Once again, we would like to thank the Reviewer for the insightful suggestion, which we believe resulted in a much-improved manuscript that may be acceptable for publication.
Andrea Agnes Molnar, MD, PhD corresponding author
|
||
Reviewer 2 Report
Comments and Suggestions for Authors
This is a very extensive and relevant review of the interrelation between cancer and heart disease. It offers an overview of several mechanisms, for which the authors are commended. Especially the potential oncogenesis in heart failure and heart disease in cancer survivors are clinically important.
Tabulating the results of the most important series and meta-analyses concerning the different classes of chemotherautic agents could facilitate the interpretation of this manuscript.
Because of the extensive character, a search strategy with MeSH terms, inclusion and exclusion criteria for manuscripts seems not feasible. This carries the risk for some bias and should be acknowledged as a limitation.
